# Microbial Communities and Physiochemical Properties of Four Distinctive Traditionally Fermented Vegetables from North China and Their Influence on Quality and Safety

**DOI:** 10.3390/foods11010021

**Published:** 2021-12-22

**Authors:** Tingting Liang, Xinqiang Xie, Lei Wu, Longyan Li, Haixin Li, Yu Xi, Ying Feng, Liang Xue, Moutong Chen, Xuefeng Chen, Jumei Zhang, Yu Ding, Qingping Wu

**Affiliations:** 1School of Food and Biological Engineering, Shaanxi University of Science and Technology, Xi’an 710021, China; gdim_liangtt@outlook.com (T.L.); wuleigdim@163.com (L.W.); chenxf@sust.edu.cn (X.C.); 2Guangdong Provincial Key Laboratory of Microbial Safety and Health, State Key Laboratory of Applied Microbiology Southern China, Guangdong Institute of Microbiology, Guangdong Academy of Sciences, Guangzhou 510070, China; woshixinqiang@126.com (X.X.); 18868006204@163.com (L.L.); lihaixinscut@163.com (H.L.); xiyu_0604@163.com (Y.X.); fengying717@163.com (Y.F.); xueliang@gdim.cn (L.X.); cmtoon@hotmail.com (M.C.); zhangjm926@126.com (J.Z.); 3College of Life Sciences, Yan’an University, Yan’an 716000, China; 4Department of Food Science & Technology, Institute of Food Safety and Nutrition, Jinan University, Huangpu Avenue 601, Guangzhou 510632, China

**Keywords:** fermented vegetables, microbial communities, physiochemical properties, quality, safety

## Abstract

The bacterial communities and physicochemical characteristics of four types of extremely distinctive traditionally fermented vegetables (pickled pepper (PP), pickled Brassica napobrassica (PBN), salted flowers of wild chives (SFWC), and pickled cucumber (PC)) were identified and compared from north China. *Lactobacillus* was the main bacterial genus in PP and PBN samples, with *Oceanobacillus* only being observed in PBN. The predominant genus in SFWC was *Weissella*, while in PC they were were *Carnimonas* and *Salinivibrio*. At the species level, *Companilactobacillus ginsenosidimutans, Fructilactobacillus fructivorans*, and *Arcobacter marinus* were abundant in PP and PBN. *Levilactobacillus brevis and Companilactobacillus alimentarius* were enriched in PP, and *L. acetotolerans*, *Ligilactobacillus acidipiscis* and *Pediococcus parvulus* were observed in PBN. *Weissella cibaria* and *Kosakonia cowanii* were abundant in SFWC. Moreover, tartaric acid was the most physicochemical factor influencing microbial composition, followed by malic acid, titratable acidity (TA), and lactic acid. Furthermore, functional analysis demonstrated that the most genes of the bacterial profiles correlated with carbohydrate metabolism. However, some foodborne pathogens were existed, such as *Staphylococcus* and *Arcobacter marinus*. The results of this study provide detailed insight into the relationship between the bacterial communities and physicochemical indices of fermented vegetables, and may improve the quality and safety of traditional Chinese fermented vegetables.

## 1. Introduction

In China, fermented vegetable pickles have a long history of more than 3000 years, dating back to the Zhou Dynasty [1,2]. To date, most people prepare fermented vegetables using traditional methods under uncontrolled fermentation conditions, and by introducing a number of microbials during the fermentation process [3,4]. Previous studies have demonstrated that microbial communities play an essential role in the quality properties of traditionally fermented foods [5]. However, the bacterial composition, structure, and population of fermented vegetables are influenced by many factors, such as geographic and climatic conditions, fermentation temperatures and times, manufacturing procedures, and vegetable type [6,7].

Numerous studies have implied that there are big differences in the microbial communities of different types of vegetables [1,2]. Recently, with the development of next-generation sequencing (NGS), an increasing number of studies have elucidated the complex microbial communities of various fermented foods. Chao et al. suggested that lactic acid bacteria (LAB) may be the core bacteria used for the production of fermented vegetables [8]. Nevertheless, the enriched LAB microbiota differed significantly among different kinds of fermented vegetables [9]. According to the report of Zhanggen et al., *Lactobacillus* and *Pediococcus* are the dominant bacterial genus in Suancai, which includes species such as *Pediococcus parvulus*, *Loigolactobacillus coryniformis, Lactiplantibacillus pentosus, and Levilactobacillus parabrevis* [10]. *Lactobacillus*, *Leuconostoc,* and *Weissella* are enriched in kimchi [11]. *Enterococcus faecalis*, *Lactobacillus delbrueckii* subsp. *lactis*, *Leuconostoc mesenteroides*, *Lactiplantibacillus plantarum subsp. plantarum, Lacticaseibacillus casei, and Lacticaseibacillus zeae* are more abundant in the Sichuan paocai [12]. Worryingly, *Escherichia coli*, *Salmonella*, *Campylobacter*, *Listeria monocytogenes,* and *Staphylococcus aureus* were observed in several fermented foods [13]. *Bacillus cereus* and *Escherichia coli* were detected in kinema, a fermented soya bean food [14]. Therefore, some measures to control the food-borne pathogens in traditional fermented vegetables should be taken.

Pickled pepper (PP), pickled *Brassica napobrassica* (PBN), salted flowers of wild chive (SFWC), and pickled cucumber (PC) are four types of extremely distinctive traditionally fermented vegetables that have been consumed as seasoned auxiliary foods for a long history in north China, where the weather is extremely cold during the winter, leading to local people being particularly fond of fermented vegetables. In this manner, they are able to consume vegetables during the cold and long winter. Therefore, research on the quality and safety of these traditionally fermented vegetables is imperative. However, studies on these traditionally fermented vegetables have mainly focused on their flavor and nutrition [15,16]. Moreover, there are still knowledge gaps in terms of a comparison of the overall differences of these fermented vegetables, such as their taste and flavor. In particular, the differences among the bacterial communities of these four fermented vegetables remain unknown. Additionally, the complex correlations between the key microbial communities and the physicochemical characteristics of these traditionally fermented vegetables remain unclear.

In this study, we used 16S rRNA amplicon sequencing to evaluate and compare the microbial communities of four traditionally fermented vegetables that were collected from different regions of north China. The relationship between bacterial diversity and physiochemical indices was investigated, and foodborne pathogens were also identified. The results of this study may be helpful for selecting optimal and appropriately fermented vegetables and for the industrial production of high-quality fermented vegetables in north China. It also helps to build on indigenous knowledge on food processing and its impact on product quality and safety.

## 2. Materials and Methods

### 2.1. Sample Collection

All 18 fermented vegetable samples (four PPs, four PBNs, five SFWCs and five PCs) were collected from different areas in Harbin and Inner Mongolia in north China. All samples were collected in August 2019, with the temperature in this season being 15–30 °C in these two areas, with a significant temperature difference between day and night. Most samples were collected from peasant households and farmer’s markets. The manufacturing process for these four traditionally fermented vegetables were also recorded. For PP production, the fresh peppers were washed, and a small cut was made on them. Then, the peppers were placed into a jar, in which cooled boiling water with 0.5–3.5% salt was poured into the jar, and the jar was pressed with stones. The spontaneous fermentation in the jar lasted for approximately 10 days. The PBN production involved washing and cutting *Brassica napobrassica* into thin strips, and placing it into a jar. Then, salt was added, and a weight was placed on top of the Brassica napobrassica. The spontaneous fermentation in the jar lasted approximately 3–5 days. For SFWC production, wild leek flower buds were washed and ground with a mixer machine. Then, some salt was added, which was mixed well with the buds. Next, the buds were placed into a jar, and spontaneous fermentation lasted for two days. For PC production, the cucumbers were washed and cut into thin slices. Then, some salt was added, mixed well with the cucumber slices, and the mixture was allowed to marinate for 1 h until the water came out. Next, the water was squeezed out of the cucumbers, which were then placed into a jar, and refrigerated them for approximately three days.

The collected samples were placed into sealed aseptic bags, transferred into a foam box filled with dry ice, and immediately transported to the laboratory. The samples were placed in a refrigerator at −20 °C before analysis.

### 2.2. Physicochemical Analysis

The pH and titratable acidity (TA) of the samples were determined based on the methods described in a previous study [17,18], and TA was titrated using 0.1 mol/L NaOH with 1% phenolphthalein-ethanol solution as indicator, until a light pink color appeared. The contents of organic acids (lactic, tartaric, malic, and acetic acids) were determined with HPLC, as described in a previous study [19].

### 2.3. DNA Eextraction and Polymerase Chain Reaction (PCR) Amplification

Microbial DNA from 18 fermented vegetable samples was extracted using the E.Z.N.A.^®^ soil DNA Kit (Omega Bio-tek, Norcross, GA, USA), according to the manufacturer’s instructions. The concentration and purity of genomic DNA were determined using a NanoDrop 2000 UV-Vis spectrophotometer (Thermo Fisher Scientific, Waltham, MA, USA), and DNA quality was assessed using 1% agarose gel electrophoresis. The V3-V4 hypervariable regions of bacterial 16S rRNA genes were amplified with primers 338F (5′- ACTCCTACGGGAGGCAGCAG-3′) and 806R (5′-GGACTACHVGGGTWTCTAAT-3′) using a thermocycler (GeneAmp 9700; Applied Biosystems, Waltham, MA, USA). The PCR reaction conditions were as follows: 3 min of denaturation at 95 °C, 27 cycles of 30 s of denaturation at 95 °C, 30 s of annealing at 55 °C, and 45 s of elongation at 72 °C, with a final extension at 72 °C for 10 min. The PCR mixture contained 4 μL 5 × FastPfu Buffer, 2 μL 2.5 mM dNTPs, 0.8 μL of each primer (5 μM), 0.4 μL FastPfu Polymerase, and 10 ng template DNA. Amplicons were extracted from a 2% agarose gel and further purified using the AxyPrep DNA Gel Extraction Kit (Axygen Biosciences, Union City, CA, USA) and quantified using QuantiFluor™-ST (Promega, Durham, NC, USA), according to the manufacturer’s instructions. All PCR amplifications were repeated three times.

### 2.4. Illumina MiSeq Sequencing

The purified amplicons were pooled in equimolar concentrations, and were paired-end sequenced using an Illumina MiSeq PE300 platform/NovaSeq PE250 platform (Illumina, San Diego, CA, USA), according to the standard protocols by Majorbio Bio-Pharm Technology Co. Ltd. (Shanghai, China). The raw reads were deposited in the NCBI Sequence Read Archive database (Accession Number: PRJNA776401).

### 2.5. Sequencing Data Processing

The raw 16S rRNA gene sequencing reads were de-multiplexed, quality-filtered using fastp version 0.20.0 [20], and merged using FLASH version 1.2.7 [21] with the following criteria: (i) the 300 bp reads were truncated at any sites receiving an average quality score < 20 over a 50 bp sliding window, and truncated reads shorter than 50 bp were discarded. Reads containing ambiguous characters were also discarded; (ii) only overlapping sequences longer than 10 bp were assembled according to their overlapped sequence. The maximum mismatch ratio of the overlap region was 0.2. Reads that could not be assembled were discarded; (iii) samples were distinguished according to the barcode and primers, the sequence direction was adjusted, and the exact barcode was matched using a two-nucleotide mismatch in primer matching.

Operational taxonomic units (OTUs) with a 97% similarity cut-off were clustered using UPARSE (version 7.1) [22], and chimeric sequences were identified and removed. The taxonomy of each OTU representative sequence was analyzed using the RDP Classifier [23] against the 16S rRNA database (Silva v138) with a 0.7 confidence threshold.

### 2.6. Statistical Analysis

The differences in physicochemical indicators among all groups were analyzed using a *t*-test. A *p*-value of less than 0.05 was considered significant. All data are described as mean ± standard deviation (SD). The graphs were generated using GraphPad Prism 7 (GraphPad Software, Inc., La Jolla, CA, USA). Rarefaction analysis and alpha diversities were performed using Mothur (version v.1.30.1, http://www.mothur.org) (accessed on 10 January 2021). Bray Curtis similarity clustering analysis was performed using the R package (R 3.0.2, http://cran.r-project.org/) (accessed on 10 January 2021). A Mann–Whitney *U* test was used to assess the different taxonomies of the bacterial community. Spearman’s correlation analysis was responsible for determining the relationship between the bacterial composition and the physicochemical indicators (pH, TA, lactic, acetic, malic, and tartaric acids). Functional analyses of the bacterial community in all fermented vegetables were performed using Phylogenetic Investigation of Communities by Reconstruction of Unobserved States (PICRUST) 1.0.0, according to the Greengene 16S rRNA gene dataset.

## 3. Results

### 3.1. Physicochemical Properties of the Four Traditionally Fermented Vegetables

As displayed in Figure 1, the pH values of the four traditionally fermented vegetable samples ranged from 3.33 to 7.10, with PBN samples having the lowest pH value, and significantly lowered than SFWC samples. The TA (2.18 ± 0.79 g/L), lactic acid (14.04 ± 1.19 mg/g), and malic acid (9.96 ± 1.98 mg/g) were significantly higher in the PBN samples than in PP samples, and the TA in the PBN samples was significantly higher than SFWC and PC samples (Figure 1B,C,E). The PBN samples had higher acetic acid levels (507.72 ± 15.91 mg/g) (Figure 1D). The tartaric acid content was higher in the SFWC samples (34.68 ± 3.75 mg/g) than in the PC samples (Figure 1F). In total, the PP and PC samples had lower organic acid levels.

### 3.2. Diversity Index Comparison of the Four Traditionally Fermented Vegetables

PP, PBN, SFWC and PC are four kinds of extremely distinctive traditionally fermented vegetables; however, they possess different physiochemical characteristics, i.e., acidity, taste, and flavor, which are influenced by the specific bacterial community. Therefore, we compared the diversity index comparison of the four traditionally fermented vegetables. From Figure 2A, we can see that the convergence of all rarefaction curves based on OTUs at 97% similarity stabilized for all samples, and the coverage indexes ranged from 0.994 to 1 (Figure 2B), indicating that our sequencing depth was sufficient to analyze the microbial community in all samples. Table 1 shows the sequence data, OTU numbers, and the indices of alpha-diversity, which were 53,610 per sample on average, and a total of 964,996 reads were obtained for all samples. The mean OTUs of the PP, PBN, SFWC, and PC samples were 143, 343, 197, and 160, respectively. The Shannon, Chao, Ace, and Simpson indices varied in all samples. The Chao and Ace indices were significantly higher in the PBN samples than in the PP samples (Figure 2D,E). The SFWC samples had the lowest Shannon index, and the highest Simpson index (Figure 2C,F), indicating that the bacterial richness and bacterial community diversity were significantly higher in the PBN samples than in the other three groups.

### 3.3. Bacterial Community Comparisons

Principal component analysis (PCA) and non-metric multidimensional scaling (NMDS) were used to compare bacterial composition within these four groups. The PCA results suggested that there were significant differences between the SFWC and other three group samples (Figure 3A). The NMDS analysis in consistent with PCA results. From Figure 3B, we can see that SFWC samples were apparently separated from the other three group samples, and other three group samples overlapped with one another, implying that the bacterial composition structure of SFWC samples was extremely different to other three samples. Meanwhile, as shown in Figure 3C, the bacterial communities of all samples were classified into two OTU types based on their composition.

The common and unique OTUs and genera among the four group samples are described in a Venn diagram (Figure 3D,E). At the OTU level, 108 OTUs were common among different samples. The PBN samples had more unique OTUs than the other three groups (404) (Figure 3D). In addition, the PBN samples showed a much greater variation of unique genera than the other three groups (Figure 3E), suggesting that these unique genera played a dominant role in PBN samples.

### 3.4. Bacterial Profiles of the Four Traditionally Fermented Vegetables

The composition of the bacterial communities in all samples depending on OTUs at 97% similarity and bacterial annotation are shown in Figure 4. Overall, a total of 28 phyla, 624 genera, and 1030 species were identified in all samples.

At the phylum level, Cyanobacteria, Firmicutes, and Proteobacteria were the predominant phyla in all fermented vegetables and accounted for over 90% of the annotated reads, followed by Epsilonbacteraeota, Actinobacteria, Bacteroidetes and Halanaerobiaeota (Figure 4A). The abundance of Cyanobacteria was relatively higher in SFWC (82.6%), followed by PP (40.6%), PC (38.1%), and PBN (3.8%). The PBN samples had the highest relative abundance of Firmicutes, accounting for 66.8%, followed by PP (37.0%), SFWC (8.7%), and PC (6.9%). Proteobacteria, the third predominant phylum, accounted for 51.2%, 22.1%, 16.6%, and 7.9% in PC, PBN, PP, and SFWC samples, respectively.

At the genus level, *Lactobacillus*, *Carnimonas*, *Salinivibrio*, *Weissella*, *Arcobacter*, and *Halomonas* were the most representative genera in all samples (Figure 4B). *Lactobacillus* was the most dominant bacterial genus. *Lactobacillus* (Firmicutes) is important in the fermentation process of vegetables. It is noteworthy that *Lactobacillus* was the major genus in PBN (49.6%), PP (34.6%), and PC (5.1%). However, *Lactobacillus* was not detected in SFWC samples, while *Weissella* was the predominant genus in SFWC samples (7.7%). *Carnimonas*, *Salinivibrio*, and *Terasakiispira* were also highly abundant in PC samples (29.9%, 8.3%, and 4.1%, respectively). In addition, *Salinivibrio*, *Weissella*, *Arcobacter*, *Halomonas*, and *Terasakiispira* were more abundant in PBN samples. Moreover, *Pediococcus*, *Staphylococcus*, and *Cobetia* were enriched in PBN samples. Interestingly, *Oceanobacillus*, a bacterium that is rarely found in fermented vegetables, was only observed in the PBN samples in this study, respectively.

At the species level, *Companilactobacillus ginsenosidimutans, Fructilactobacillus fructivorans, A. marinus, W. cibaria, L. acetotolerans, Ligilactobacillus acidipiscis, Lactiplantibacillus plantarum* subsp. *plantarum, Levilactobacillus brevis, Pediococcus parvulus,* and *Companilactobacillus alimentarius* were the top 10 identified species (Figure 4C). *Companilactobacillus ginsenosidimutans, Fructilactobacillus fructivorans, Acrobacter marinus, L. acetotolerans, Ligilactobacillus acidipiscis, Lactiplantibacillus plantarum* subsp. *Plantarum,* and *Levilactobacillus brevis* were more abundant in the PP, and PBN samples. Moreover, *L. acetotolerans, Ligilactobacillus acidipiscis, Pediococcus parvulus, Staphylococcus saprophyticus,* and *Oceanobacillus oncorhynchi* subsp. *incaldanensis* were abundant in PBN samples. *W. cibaria* was markedly more abundant in SFWC samples (7.7%), followed by PC (0.08%), PBN (2.8%), and PP (0.2%). However, unclassified species belonging to the *Carnimonas, Salinivibrio,* and *Terasakiispira* genera were highly abundant in SFWC and PC samples.

In addition, in Figure 4B,C, the bar of four kinds of sample are different. It was found that the SFWC samples had a higher relative abundance of norank_f_norank_o_*Chloroplast*, accounting for 82.6%, followed by PC (38.1%), PP (30.3%), and PBN (3.8%); however, this species belongs to *Chloroplast*, not the bacteria, thus we removed it at the genus level; Meanwhile, we found that the SFWC samples had the higher relative abundance of *Allium sativum garlic*, accounting for 81.4%, followed by PP (1.0 %), PC (0.6%), and PBN (0.2%), and the PC samples had a higher relative abundance of unclassified_g_norank_f_norank_o_*Chloroplast*, accounting for 37.3%, followed by PP (29.2 %), PBN (3.6%), and SFWC (1.2%) at the species level; we found that these two species were not bacteria, so we removed them.

### 3.5. Differential Bacteria in the Four Traditionally Fermented Vegetables

According to the relative abundance of the microbial composition, a significant difference analysis of microbial communities in different groups was carried out using the Linear Discriminant Analysis Effect Size (LEfSe) analysis. As shown in Figure 5, 13 genus level bacterial were observed to be differentially enriched in these four traditional fermented vegetables, and 10 of them (including *Lactobacillus*, *Macrococcus*, *Halomonas*, *Pediococcus*, *Oceanobacillus*, *Staphylococcus*, *Tetragenococcus*, *Ralstonia*, *Idiomarina*, and *Marinilactibacillus*) were enriched in PBN samples, while only *Marinobacter*, *Weissella*, *Terasakiispira* was abundant in PP, SFWC, and PC samples, respectively.

### 3.6. Correlation of Bacterial Communities and the Physicochemical Indices of the Four Traditionally Fermented Vegetables

The results of CCA analysis (Figure 6) suggested that there was a strong correlation between physicochemical indexes and bacterial composition structure. The first axis (horizontal) was positively related to the contents of TA, lactic, and malic acid, while the second axis (vertical) was strongly correlated with tartaric acid. Tartaric acid had the greatest effect on the bacterial composition, followed by malic acid, TA, and lactic acid. Nevertheless, pH and acetic acid did not influence the bacterial composition. According to the Mantel test analysis, tartaric acid (r = 0.7033, *p* = 0.001) had the strongest correlation with the microbial abundance, followed by malic acid (r = 0.5939, *p* = 0.004), TA (r = 0.5582, *p* = 0.001), and lactic acid (r = 0.5192, *p* = 0.002). Nevertheless, pH (r = 0.0199, *p* = 0.777) and acetic acid (r = 0.2576, *p* = 0.07) were weakly correlated with the microbial composition. In addition, it was observed that the explanatory power of the two main CCAs in the CCA combined was only 16.3%, which was a limit, meanwhile, the relationship between the specific bacterial and physicochemical indexes need be further studied.

As shown in Figure 7, the abundances of *A. marinus, L. acetotolerans, Levilactobacillus brevis, Companilactobacillus ginsenosidimutans, Lactiplantibacillus plantarum subsp. plantarum, Paralactobacillus selangorensis, O. oncorhynchi subsp. incaldanensis,* and *P. parvulus* were positively correlated with TA (*p* < 0.05). *Lactobacillus acetotolerans,* and *O. oncorhynchi* subsp. *incaldanensis* were positively associated with the contents of lactic and acetic acid (*p* < 0.05). However, *A. marinus, L. acetotolerans, Ligilactobacillus acidipiscis, Companilactobacillus alimentarius, Levilactobacillus brevis, Companilactobacillus ginsenosidimutans, Lactiplantibacillus plantarum subsp. plantarum, Limosilactobacillus reuteri,* and *P. parvulus* were negatively correlated with tartaric acid (*p* < 0.05). Moreover, *Fructilactobacillus fructivorans* and *O. oncorhynchi* subsp. *incaldanensis* were negatively and positively correlated with malic acid, respectively, which was in accordance with a higher abundance of these bacterial in the PBN group.

### 3.7. Predicted Functions of Metabolic

PICRUST was used to predict functions of microbial communities in the four traditionally fermented vegetable samples. Bacterial related genes were observed involved in metabolism, genetic information processing, environmental information processing, cellular processes, human diseases, and organism systems in all groups (Figure 8A). Metabolism-related genes were enriched in all groups, suggesting that metabolism plays a key role in microbial communities. At level 2, carbohydrate metabolism exhibited a higher level in all groups (Figure 8B). Moreover, amino acid metabolism, energy metabolism, cofactor and vitamin metabolism, lipid metabolism, xenobiotic biodegradation and metabolism, glycan biosynthesis and metabolism, and terpenoid and polyketide metabolism showed significant differences among all groups. Carbohydrate metabolism was enriched pathways in the PBN samples; it mainly included pyruvate metabolism, glycolysis/gluconeogenesis, and amino sugar and nucleotide sugar metabolism, as well as starch and sucrose metabolisms (Figure 8C). All four pathways also exhibited higher proportion in the PBN samples.

## 4. Discussion

In this study, we investigated and compared the bacterial communities and physicochemical characteristics of the PP, PBN, SFWC, and PC samples using high-throughput sequencing technology and HPLC. Furthermore, we have attempted to correlate the relationship between the bacterial profiles and physicochemical indices (organic acids) of the fermented vegetable samples.

The dominant bacteria in the PP, PBN, SFWC, and PC samples may vary greatly due to the types of vegetables and the production process. *Lactobacillus* was the most dominant genus in the PP, and PBN samples, which is consistent with previous studies on Sichuan paocai [3], Guangyuan Suancai [24], and Yancai [25]. However, *Weissella* was the predominant genera in the SFWC samples, and *Carnimonas* was detected at very high levels in the PC samples. This difference could be due to the fermentation temperature and time, fermentation process and equipment, geographical distribution, and vegetable varieties used [4,26]. Nevertheless, the results of a previous study showed that *Weissella* was also highly abundant in fermented vegetables [27,28], which was in accordance with the results of our study. In addition to *Lactobacillus* and *Weissella*, other genera found to be prevalent in PBN samples were *Carnimonas*, *Salinivibrio*, *Arcobacter*, *Halomonas,* and *Terasakiispira*. These results are quite different from the results of previous studies on other fermented vegetables. This difference involves the fermentation of materials themselves, the manufacturing process, and the equipment. Zhou et al. reported that *Lactobacillus*, *Leuconostoc*, *Enterobacter*, and *Accumulibacter* were the predominant bacterial genera in Suan-cai samples [29]. Our results also demonstrated that *Oceanobacillus* was abundant in PBN samples. These results indicate that the bacterial communities in PBN are distinct from those in other fermented vegetables. Previous studies have reported that *Oceanobacillus* can be isolated from fermented shrimp paste in Thailand [30], and also isolated from a marine solar saltern [31], so we can conclude that *Oceanobacillus* was abundant in PBN samples probably derived from used salt, and this should be studied further.

Previous studies have suggested that there is a microbial safety risk and potential for spoilage and deterioration in homemade fermented vegetables [1,3]. It is worth noting that *Staphylococcus*, an opportunistic pathogen that can cause septicemia, endocarditis, pneumonia, and meningitis by producing a variety of exotoxins and enzymes [32], was relatively highly abundant in the PBN samples, implying that PBN samples may be accidentally contaminated in the production process owing to the unhygienic production environment (Figure 9A). Meanwhile, *Arcobacter marinus* enriched PP samples (Figure 9B); it has been isolated from a water canal contaminated with urban sewage [33]. Thus, it is crucial to take appropriate measures to inhibit potentially pathogenic bacteria and other bacterial contaminants in fermented vegetables.

The results also suggested significant differences in the bacterial composition of the four traditionally fermented vegetables examined. A previous study reported that *Lactiplantibacillus plantarum* subsp. *plantarum* was the predominant *Lactobacillus* species in traditionally fermented vegetables [12]. Nevertheless, this species was relatively lower abundant in our study. *Companilactobacillus ginsenosidimutans* was significantly abundant in the PP samples, which has also been found in kimchi, a Korean fermented food, as it can biotransform ginsenosides and improve the taste of foods [34]. The relative abundances of *L. acetotolerans* and *Ligilactobacillus acidipiscis* were much higher in the PBN samples than in the other four traditionally fermented vegetables. *L. acetotolerans*, which is tolerant to acetic acid bacteria [35], was also found to be abundant in Chongqing radish paocai [36], and can produce sorts of organic acids, including lactic, acetic acid, and et al. [37]. *Ligilactobacillus acidipiscis* has also been isolated from fermented fish in Thailand [38]. This species, isolated from pickles, has been reported to have pancreatic lipase inhibitory activity [39]; thus, these two bacterial species may play an important role in the flavor of PBN samples. These differences of microbial communities may be due to use different kinds of raw materials, fermentation time and temperature, and regions. Meanwhile, from the records of the manufacturing process of these four traditionally fermented vegetables, it was observed that the fermentation conditions of these four fermented vegetables were essentially the same except for the different raw materials and fermentation time, such as the bacteria that may exist on the surface of these four fresh vegetables, the bacterial composition on the surface of these four fresh vegetables can be further studied. Moreover, the PP, PBN, SFWC, and PC production lasted for approximately 10 days, 3–5 days, 2 days, and 3 days, respectively. Therefore, we speculated that the raw materials and fermentation time might be the reasons for the differences in bacterial composition of these four fermented vegetables.

Tartaric acid was the most essential physicochemical factor influencing the microbial composition based on the CCA results, followed by malic acid, TA, and lactic acid. Xiong et al. reported that TA may be the major factor affecting bacterial communities [12]. This is the first study to report that tartaric acid and malic acid affect the structure of bacterial communities in fermented vegetables. These differences between previous studies and our study may be due to the kinds of vegetables and fermentation conditions, including the fermentation temperature and fermentation facility employed [40,41]. The results also suggested that tartaric acid was positively correlated with the bacterial communities of SFWC samples, while malic acid and lactic acid were positively correlated with PBN samples. This may be due to the higher amount of malic acid in PBN and the higher amount of tartaric acid in SFWC. Thus, it is necessary to explore the influence of tartaric acid and malic acid on microbial communities in the production of SFWC and PBN samples. Additionally, lactic acid has been previously reported to be abundant in traditionally fermented vegetables [3,12], which is consistent with our results. Therefore, organic acids are important factors that affect the microbial structure in fermented vegetables.

Previous studies have reported that most Firmicutes are positively associated with TA, lactic acid, and acetic acid in fermented vegetables [1,26,42,43], which is in agreement with the findings of our study. The abundances of *L. acetotolerans, Levilactobacillus brevis, Companilactobacillus ginsenosidimutans, Lactiplantibacillus plantarum subsp. Plantarum, O. oncorhynchi subsp. Incaldanensis, and P. parvulus* were positively associated with TA, lactic acid, and malic acid in this study. However, the aforementioned Firmicutes species had a negative correlation with tartaric acid, a result which is clearly different from that of another report [3]. Therefore, the relationship between physicochemical indices (metabolites) and bacterial community composition in fermented vegetables needs to be further studied.

It was revealed that there was significant distinctive in the metabolic related genes among these four groups. Carbohydrate metabolism-related genes constituted the most abundant pathway in all groups, and were enriched in PBN samples. Meanwhile, amino acid metabolism was abundant in all samples, which has been reported to be beneficial for the production of small amino acid flavor substances [44]. Moreover, the PBN samples had a higher abundance of glucose-lipid metabolism pathways, which is in accordance with the higher level of TA, lactic acid, and malic acid.

However, there are several important limitations in this study. Firstly, using NGS by 16s rRNA is generally not reliable to obtain relative abundance results at the species level, thus metagenomics technology is essential. Meanwhile, the content of salt and the fungal diversity were important in the fermented vegetables, and no investigation was carried out. It is necessary to measure this in future studies. Moreover, this study aimed to reveal the correlation between the microbes and the physicochemical indices, thus metabolomics technology is essential. In addition, gene abundance related to putative main pathways, transcriptional studies have to be performed. Finally, our future work is to correlate the relative abundance of pathogens with counts using traditional methods to confirm whether they are cultivable or not.

## 5. Conclusions

In this study, we investigated differences in the microbial communities and physicochemical indices of the four traditionally fermented vegetables collected from different regions in north China. *Loigolactobacillus rennini* and *O. oncorhynchi* subsp. *incaldanensis* were abundant in PBN samples. *W. cibaria* was particularly abundant in SFWC samples. In addition, TA, lactic acid, and malic acid were significantly higher in the PBN samples than in the other four groups. The PBN samples had higher acetic acid levels, while tartaric acid was significantly higher in the SFWC than in the PC samples. There was a significant correlation between bacterial communities and physicochemical indices in the four traditionally fermented vegetables, and the core bacteria, including *L. acetotolerans, O. oncorhynchi subsp. incaldanensis, A. marinus, Ligilactobacillus acidipiscis, Companilactobacillus alimentarius, Levilactobacillus brevis, Companilactobacillus ginsenosidimutans, Lactiplantibacillus plantarum subsp. plantarum, Loigolactobacillus rennini, P. parvulus, and Fructilactobacillus fructivorans* contributed to the development of major organic acids. In addition, the PICRUST results suggested that most of the genes of the bacterial profiles in the four traditionally fermented vegetables were correlated with carbohydrate metabolism, which might promote the production of the flavor (e.g., organic acids).

## Figures and Tables

**Figure 1 foods-11-00021-f001:**
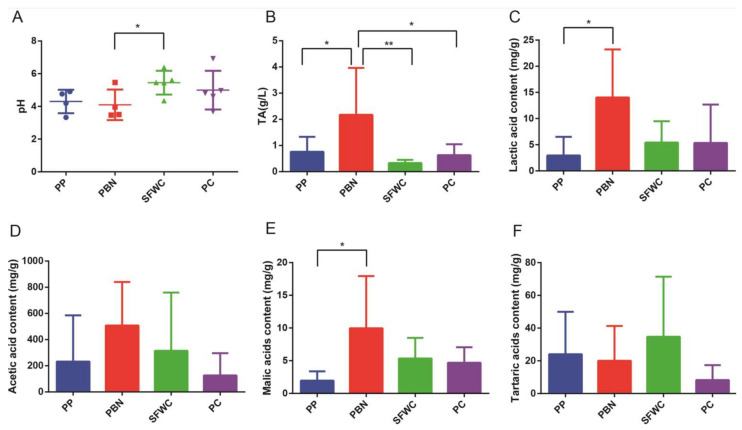
Comparison of physicochemical characteristics in four traditional fermented vegetables. (**A**) pH; (**B**) TA; (**C**) lactic acid; (**D**) acetic acid; (**E**) malic acid; (**F**) tartaric acid. * *p* < 0.05, ** *p* < 0.01.

**Figure 2 foods-11-00021-f002:**
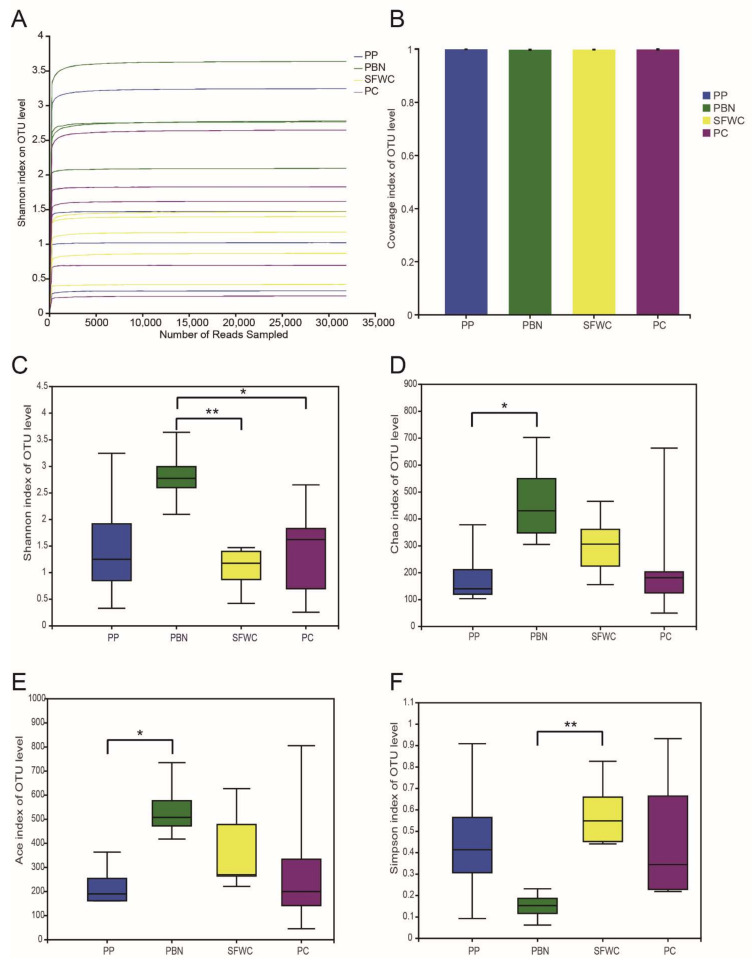
Alpha-diversity of bacterial community of four traditional fermented vegetables. (**A**) Shannon curve for each sample. (**B**) Coverage index. (**C**) Shannon index. (**D**) Chao value. (**E**) Ace index. (**F**) Simpson index. * *p* < 0.05, ** *p* < 0.01.

**Figure 3 foods-11-00021-f003:**
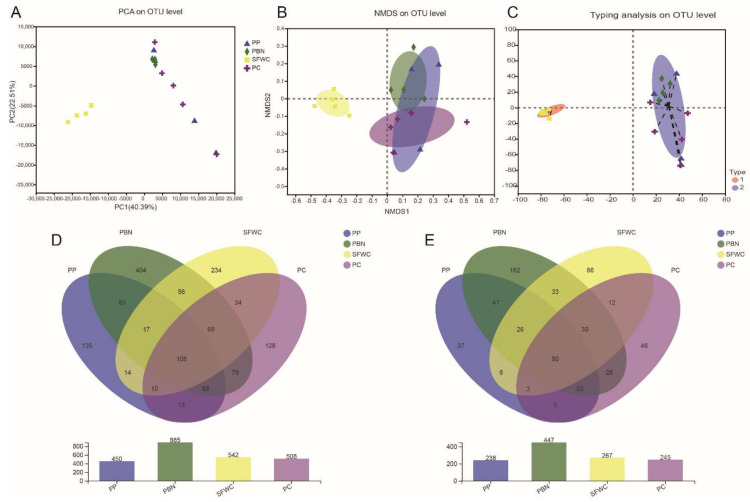
Beta diversity and Venn diagrams of bacterial community of four traditional fermented vegetables. (**A**) Principal component analysis (PCA). (**B**) nonmetric Multidimensional Scaling (NMDS). (**C**) bacterial classification plot. Venn diagrams at OTU (**D**), genus level (**E**) according to bacterial biodiversity.

**Figure 4 foods-11-00021-f004:**
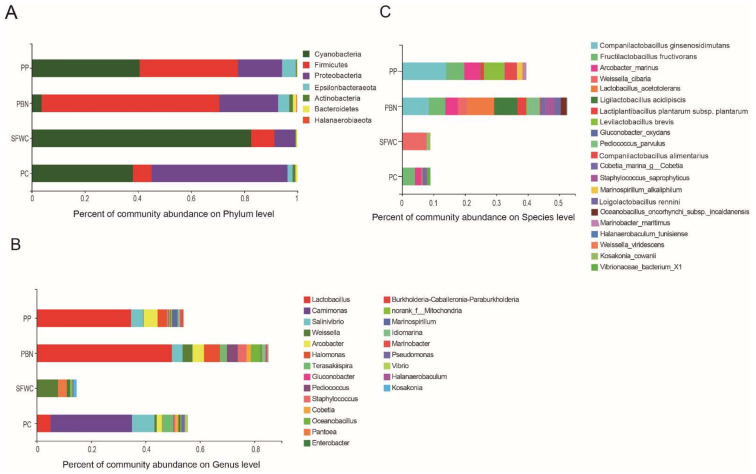
Composition of bacterial communities in four traditional fermented vegetables samples at the phylum (**A**), genus (**B**), species (**C**) level. Phyla, genera with proportions amounting to less than 1%, and unclassified species are not listed.

**Figure 5 foods-11-00021-f005:**
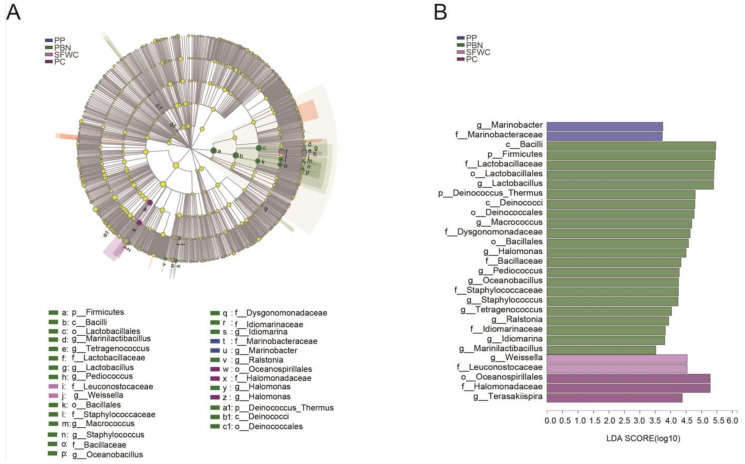
LEfSe comparison of bacterial communities among four traditional fermented vegetables samples. (**A**) Cladogram representing the abundance of those taxa in these four fermented vegetables. (**B**) Histogram of the results of the microbiota in these four fermented vegetables with a threshold value of 2.

**Figure 6 foods-11-00021-f006:**
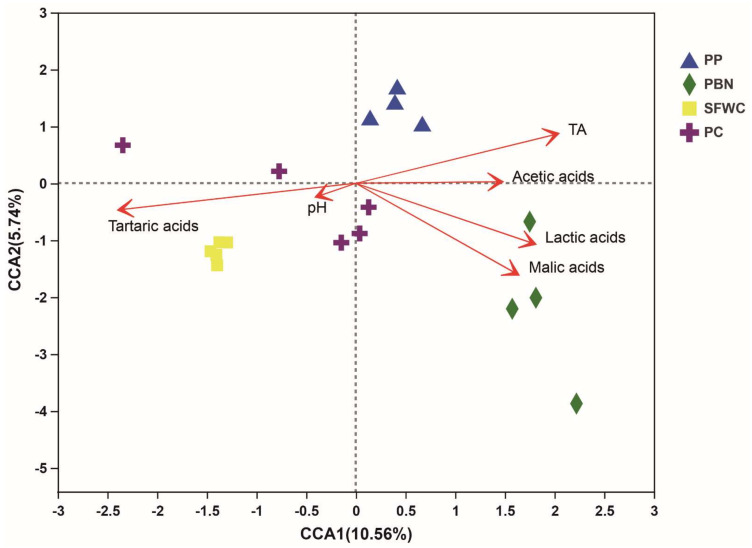
Canonical correlation analysis (CCA) of four traditional fermented vegetables samples.

**Figure 7 foods-11-00021-f007:**
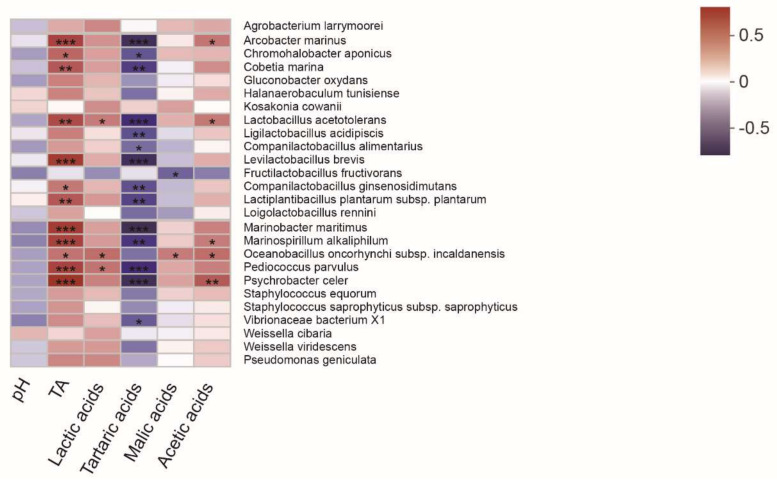
Spearman correlation analysis of the physicochemical index and dominant species in four traditional fermented vegetables. The Spearman correlation coefficient r ranges from −0.5 to 0.5; r < 0 is negative correlation, r > 0 is positive correlation. * *p* < 0.05, ** *p* < 0.01, *** *p* < 0.001.

**Figure 8 foods-11-00021-f008:**
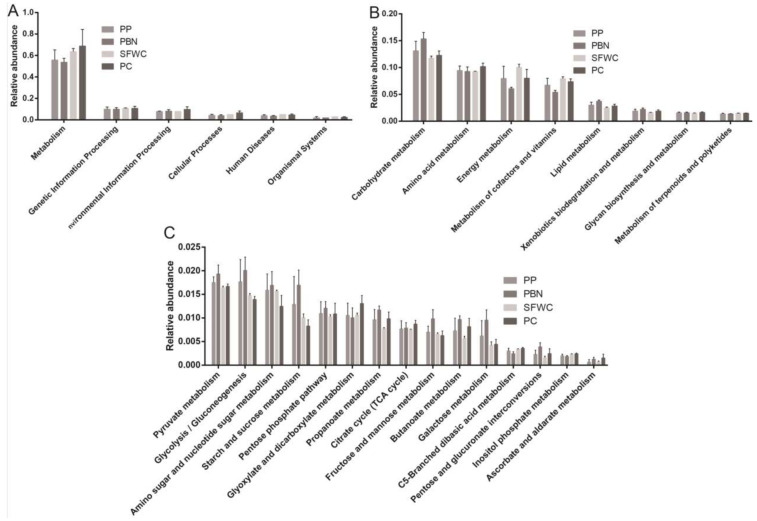
The predictive functions of the bacterial community in four traditional fermented vegetables samples. (**A**) Overall KEGG gene function statistics (Level 1). (**B**) The relative abundance of pathways in metabolism (Level 2). (**C**) The relative abundance of pathways in carbohydrate metabolism (Level 3).

**Figure 9 foods-11-00021-f009:**
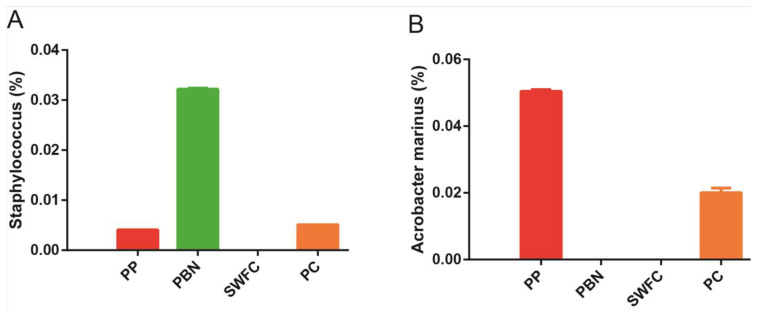
The relative abundance of foodborne pathogens in four traditional fermented vegetable samples. (**A**) *Staphylococcus*. (**B**) *Arcobacter marinus*.

**Table 1 foods-11-00021-t001:** Sequence information and diversity values of four traditional fermented vegetables in north China.

	Sample_Name	Seq_Num	Average Length (bp)	OTU	Shannon	Simpson	Ace	Chao1	Coverage
PP	PP_83	50,570	428	109	1.472	0.377	216.499	154.556	0.999
	PP_254	48,552	422	322	3.241	0.092	362.423	377.313	0.998
	PP_263	56,438	406	83	0.324	0.908	160.771	124.333	0.999
	PP_318	62,935	415	59	1.019	0.448	159.878	102.875	0.999
PBN	PBN_81	54,631	424	452	3.635	0.061	489.033	497.643	0.998
	PBN_147	51,414	427	245	2.762	0.133	416.913	361.000	0.997
	PBN_198	61,650	427	535	2.774	0.230	733.741	701.693	0.994
	PBN_260	48,502	428	141	2.090	0.171	523.611	304.235	0.998
SFWC	SFWC_118	43,810	412	158	1.395	0.440	219.884	223.900	0.998
	SFWC_120	46,411	409	251	1.171	0.547	476.835	360.306	0.997
	SFWC_238	46,848	414	287	1.465	0.451	625.914	464.349	0.996
	SFWC_239	36,547	410	197	0.865	0.658	268.803	305.043	0.998
	SFWC_257	61,458	407	94	0.416	0.826	262.913	154.882	0.999
PC	PC_132	53,280	421	86	1.825	0.217	140.644	123.800	0.999
	PC_133	50,564	424	142	1.616	0.343	198.557	180.607	0.999
	PC_192	67,831	429	34	0.693	0.663	44.481	49.000	1.000
	PC_237	63,784	416	419	2.644	0.227	804.149	662.019	0.995
	PC_269	59,771	406	119	0.249	0.931	333.021	202.182	0.998

## Data Availability

16S amplicon data from these four fermented vegetables: http://www.ncbi.nlm.nih.gov/bioproject/776401 (accessed on 29 October 2021), Submission ID: SUB10592355, BioProject ID: PRJNA776401.

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
