# Peer review of "Microbial Communities and Physiochemical Properties of Four Distinctive Traditionally Fermented Vegetables from North China and Their Influence on Quality and Safety"

_foods, 2021, doi:10.3390/foods11010021_

Round 1
Reviewer 1 Report
The authors present the relationship between the bacterial communities and physicochemical indices of Chinese fermented vegetables.
In my opinion, this manuscript is well designed, well written and it presents important data for the industrial production of high-quality fermented vegetables in China. Moreover, results will allow to have knowledge on food processing and its impact on product quality and safety.
However, some specific points need to be clarified:
Reclassification of the genus Lactobacillus, proposed in 2020 by Zheng et al., into 25 genera including the emended genus Lactobacillu (the Lactobacillus delbrueckii group) and 23 novel genera. As example, Levilactobacillus parabrevis is the name of Lactobaillus parabrevis (line 54). Please check the manuscript for the new name of genus.
Introduction:
Have any studies been investigated on the fungal community of fermented vegetables in China ?
Are they any studies on correlations between the key microbial communities and the physicochemical characteristics of fermented vegetables such as fermented soya bean foods ?
Please, explain why fungal diversity was not studied in your publication.
Materials and methods:
Lines 88-101: Please, add some details about the manufacturing process.
for PP and SWC production, what is the salt level ?
What is the temperature of fermentation ?
Results and discussion:
Fig 1 : authors said that « the PP and PC samples had lower organic acid levels». However, the pH values were different. How do you explain that ?
Lines 238-245 + lines 27_-286 : Please, use « italic » for bacterial names.
Line 321 : this difference could be due to the fermentation temperature and time, fermentation process and equipment, geographical distribution, and vegetable varieties used.
Do you have informations on bacterial diversity on raw vegetable varieties used ?
Author Response
Dear Reviewers,
Thank you for offering us an opportunity to improve the manuscript. Please find uploaded the revised manuscript entitled “Microbial Communities and Physiochemical Properties of four Distinctive Traditionally Fermented Vegetables from North China and Its Influence on Quality and Safety (Manuscript ID foods-1464675)” by Tingting Liang, Xinqiang Xie, Lei Wu, Longyan Li, Haixin Li, Yu Xi, Ying Feng, Liang Xue, Moutong Chen, Xuefeng Chen, Jumei Zhang, Yu Ding, and Qingping Wu, originally submitted on Oct 30, 2021.
We are grateful for all efforts by the reviewers to further improve our manuscript and agree with all their comments. We revised our manuscript according to the comments of the reviewers. An itemized list of changes show the responses to each comment. Changes to the manuscript were made in RED.
We hope the manuscript in the present form meets the requirements of Foods.
Yours sincerely,
Qingping Wu, PhD
Guang zhou, China, Nov. 28, 2021
E-mail address: wuqp203@163.com

Reviewer 2 Report
This article compares the microbial species and basic components of different types of fermented vegetables, it is of reference value. However, the basic ingredients and microbial species of fermented vegetables vary greatly due to different raw materials and production processes. This article ignores the effect of this part in the comparison of microbial species, which may lead to wrong conclusions. It is recommended that the statistical analysis and discussion be revisited after considering these factor.
#1. In the P4 Line 148, 0.7 was used as the confidence threshold, but there is no reference about why 0.7 was used.
#2. In the P4 Line 167-169, it showed that the TA, lactic acid, and malic acid were significantly higher in the PBN samples than 168 in the other four groups. But in Fig. 1B and 1C, the * was different. Is the asterisk labeled correctly in Fig. 1?
#3. There is no *** marked in the Fig. 1, please modified the footnote of Fig. 1.
#4. The Fig. 2B, Fig. 3B, and Fig. 4C should not be a screenshot.
#5. Fig. 2 showed the Diversity Index Comparison of the Four Traditionally Fermented Vegetables. Since the four products are very different from each other in terms of raw materials and manufacturing processes, please cite the reference to explain why the analysis is still used.
#6. In Fig. 3B, the PP were similar as PBN and PC. However, the article describes that all samples were clearly classified into four groups. Please reinterpret this section
#7. In P7 Line 204-209, it showed that the SFWC samples showed a much greater variation of unique genera than the other two groups, but in Fig. 3D and 3E, SFWC samples seems to be closer to the other two groups of samples. Please explain this part of article.
#8. In Fig. 4B and 4C, the bar of 4 kinds of sample are different, does these part mean that many species and level of microorganisms have not been identified?
#9. The r in Fig. 6 is less than 0.6, but some of the p-values are less than 0.05. Please explain this phenomenon.
#10. P13 Line 325-328, the article mentions that these results are 327 quite different from the results of previous studies on other fermented vegetables. Please explain this part of article.
#11. P13 Line 330-333, it showed that Oceanobacillus was abundant in PBN samples, which has been reported to be isolated from fermented shrimp paste in Thailand. Please explain why the PBN contains a large amount of Oceanobacillus.
#12. The form of label for significant differences in Fig. 1 and Fig. 2 should be the same.
#13. The discussion in the part of discussion should be integrated with the results of the previous Figures and Table to facilitate the reader's reading. Please strengthen this part of article.
#14. P14 Line 366-368 showed that this is the first study to report that tartaric acid and malic acid affect the structure of bacterial communities in fermented vegetables. However, this may be due to the higher amount of malic acid in PBN and the higher amount of tartaric acid in SFWC. Please discuss further on this part.
#15. P15 Line 395-402 showed that 16s rRNA is generally not reliable to obtain relative abundance results at the species level. The microorganisms in different types of fermented vegetables can vary greatly due to the types of vegetables and the production process. Even for the same kind of fermented vegetables, there are great differences in the types of microorganisms depending on the raw materials and the time and place of production. But this article did not compare the microbial species of the same kind of fermented vegetable like Fig.3-5. It is reasonable to compare different types of fermented vegetables only after confirming that the microbial species in the same type of fermented vegetables are less different. Please enhance the discussion of this part or further explain this part.
Author Response

(The authors gave the same response as above.)

Round 2
Reviewer 2 Report
#5
The topic of this manuscript is to investigate the differences of microorganisms in four different kinds of fermented vegetables. Generally speaking, the material and fermentation steps during processing should be the most important factors affecting the physical and chemical properties and flavor of the product, not only the specific bacterial community in the product. Please add a comparison of the raw materials and processes for the four types of fermented vegetable samples used in this study with reference. This is to confirm that the differences in the microorganisms of the four types of fermented vegetables discussed in this study are sufficient to represent these four types of fermented vegetables. Then, this manuscript is of reference value.
#8
Please add this part of the explanation to the manuscript to avoid misunderstanding by readers.
#9
It is reasonable that the correlation coefficient has a low p value as you mentioned. However, the explanatory power of the two main CCAs in the CCA combined is only 16.3%, this means that the explanation of this Figure is not good. Please reconfirm the results of this Figure and is there a need for this Figure.
Author Response
Dear Dr Reviewer,
Thank you for offering us an opportunity to improve the manuscript. Please find uploaded the revised manuscript entitled “Microbial Communities and Physiochemical Properties of four Distinctive Traditionally Fermented Vegetables from North China and Its Influence on Quality and Safety (Manuscript ID foods-1464675)” by Tingting Liang, Xinqiang Xie, Lei Wu, Longyan Li, Haixin Li, Yu Xi, Ying Feng, Liang Xue, Moutong Chen, Xuefeng Chen, Jumei Zhang, Yu Ding, and Qingping Wu, originally submitted on Oct 30, 2021.
We are grateful for all efforts by the reviewers to further improve our manuscript and agree with all their comments. We revised our manuscript according to the comments of the reviewers. An itemized list of changes show the responses to each comment. Changes to the manuscript were made in RED.
We hope the manuscript in the present form meets the requirements of Foods.
Yours sincerely,
Qingping Wu, PhD
Guang zhou, China, Dec. 3, 2021
E-mail address: wuqp203@163.com